# Extracellular Vesicles for Dental Pulp and Periodontal Regeneration

**DOI:** 10.3390/pharmaceutics15010282

**Published:** 2023-01-14

**Authors:** Hongbin Lai, Jiaqi Li, Xiaoxing Kou, Xueli Mao, Wei Zhao, Lan Ma

**Affiliations:** 1Department of Pediatric Dentistry, Guanghua School of Stomatology, Hospital of Stomatology, Sun Yat-sen University, Guangzhou 510055, China; 2South China Center of Craniofacial Stem Cell Research and Guangdong Province Key Laboratory of Stomatology, Guanghua School and Hospital of Stomatology, Sun Yat-sen University, Guangzhou 510055, China

**Keywords:** extracellular vesicle, pulp regeneration, periodontal regeneration

## Abstract

Extracellular vesicles (EVs) are lipid bound particles derived from their original cells, which play critical roles in intercellular communication through their cargoes, including protein, lipids, and nucleic acids. According to their biogenesis and release pathway, EVs can be divided into three categories: apoptotic vesicles (ApoVs), microvesicles (MVs), and small EVs (sEVs). Recently, the role of EVs in oral disease has received close attention. In this review, the main characteristics of EVs are described, including their classification, biogenesis, biomarkers, and components. Moreover, the therapeutic mechanism of EVs in tissue regeneration is discussed. We further summarize the current status of EVs in pulp/periodontal tissue regeneration and discuss the potential mechanisms. The therapeutic potential of EVs in pulp and periodontal regeneration might involve the promotion of tissue regeneration and immunomodulatory capabilities. Furthermore, we highlight the current challenges in the translational use of EVs. This review would provide valuable insights into the potential therapeutic strategies of EVs in dental pulp and periodontal regeneration.

## 1. Introduction

Extracellular vesicles (EVs) are encapsulated particles secreted by all types of parent cells. In addition, EVs can be found in various biological fluids, such as blood, urine, and saliva [1,2]. They were firstly observed in plasma 50 years ago [3], but were regarded as cellular waste and thought insignificant for a long time. However, research in the past decade has markedly increased our knowledge about EVs in terms of their classification, characteristics, and functions. With respect to the guidelines proposed by the International Society of Extracellular Vesicles (ISEV) in 2018, EVs can be characterized by their size, origin, protein composition, and functions [4]. EVs are all enclosed by a lipid bilayer and carry complex contents such as proteins, lipids, and nucleic acids [5,6]. They participate in both physiological and pathological processes and exert similar functions to their origin cells through intercellular communication and material transmission [5,7]. The application of EVs as disease biomarkers, therapeutic targets, novel drug agents, and acellular therapeutics has attracted considerable attention [6].

Dental pulp is a vital, highly vascularized, and innervated tissue that provides several functions for teeth, such as response to bacterial insult and injury. The presence or absence of dental pulp can greatly affect the prognosis of a treated tooth [8]. However, dental pulp is susceptible to trauma and infection caused by dental caries, periodontitis, retrograde infection, or iatrogenic causes, which eventually leads to irreversible pulp injury or necrosis [9]. Endodontic therapy has been considered the primary choice for clinical treatment; however, this approach increases the fracture rate of treated teeth and the failure rate of endodontic therapy ranges from 19.1% to 25.3% [10]. Therefore, dental pulp regeneration has marked advantages in maintaining the function of teeth after pulp disease. Although cell-based therapies have been reported to be able to regenerate three-dimensional pulp tissue with blood vessels and sensory nerves [11], the application of cell-based therapies has been limited because of concerns regarding immune rejection, safety, or medical ethics [12]. Thus, there is an urgent need for alternative cell-free approaches.

Periodontitis is a highly prevalent, multifactorial, chronic inflammatory disease of the periodontium characterized by the irreversible destruction of tooth-supporting structures, including the gingiva, periodontal ligament (PDL), cementum, and alveolar bone [13]. Over 30% of adults worldwide are subjected to periodontitis, which is the main cause of adult tooth loss. However, regular treatment strategies have limited efficacy in regenerating damaged periodontal tissues [14]. Stopping disease progression and maintaining therapeutic achievements are the major goals of conventional periodontal treatments; however, complete and functional periodontal regeneration remains a clinical challenge [15].

By modifying cellular behavior via affecting target cells in a paracrine or an endocrine manner, EVs can be used to treat multiple diseases, such as cancer, neurological diseases, and cardiovascular disorders [6]. Moreover, their therapeutic potential in oral disease has become a research hotspot [16,17]. In this paper, we will systemically review the current status of EVs in the treatment of pulp and periodontal regeneration in vivo and in vitro together with mechanistic analyses.

## 2. Extracellular Vesicles

The classification of EVs is based on their biogenesis, as described below and in Figure 1, namely apoptotic vesicles (ApoVs), microvesicles (MVs), and small EVs (sEVs, also known as exosomes). The general characteristics of EVs will be introduced here, including their classification, biogenesis, biomarkers (as shown in Figure 2), and components.

### 2.1. Apoptotic Vesicles

ApoVs are phospholipid bilayer-bound vesicles generated from apoptotic cells that take part in multiple pathophysiological events [18]. ApoVs are derived from various types of cells, such as precursor cells, endothelial cells, immunocytes, and mesenchymal stem cells (MSCs) [19]. The buoyant density of ApoVs ranges from 1.118 to 1.228 g/mL, making them denser than sEVs (1.05–1.15 g/mL) and MVs (1.03–1.10 g/mL) [20,21]. Until now, the classification of ApoVs has been a matter of debate. According to their size differences, ApoVs are generally categorized as apoptotic bodies (ApoBDs), apoptotic microvesicles (ApoMVs), and apoptotic exosomes (ApoExos) [22]. ApoBDs, first recognized by Kerr et al. in 1972, represent large ApoVs ranging from 1 to 5 μm in diameter [23]. In the last decade, research has demonstrated their role in intercellular communication and their therapeutic effects on diseases [24,25]. Furthermore, ApoMVs, at 0.1–1 μm in size, were found to be physiologically different from traditional ApoBDs, possessing superior membrane integrity for molecular exchange [26,27]. Moreover, works from Hebert’s laboratory demonstrated the presence of ApoExos [28]. ApoExos have been identified to share similar physical features with exosomes including size, density, and protein contents; while Park and co-workers have shown that ApoExos contain unique protein markers, such as sphingosine-1-phosphate receptors 1 and 3 (S1PR1 and S1PR33) and can be induced by damage-associated molecular patterns (DAMPs) [29]. ApoExos are considered to be messages from a dying cell [30]. In this review, the general term ApoVs is used.

The formation of ApoVs comprises a series of well-coordinated morphological changes via a process termed apoptotic cell disassembly. The disassembly of an apoptotic cell starts with the initial membrane blebbing on the cell surface. Subsequently, apoptotic membrane protrusions occur in the form of microtubule spikes, apoptopodia, and beaded apoptopodia. Finally, these protrusions are disassembled into smaller fragments, which turn into a variety of ApoVs [31,32,33]. In general, ApoV-associated “find-me” signals help to recruit phagocytes while the exposure of “eat me” signals, like phosphatidylserine (PS), on the ApoV surface, allows them to be recognized and engulfed by phagocytes for clearance [34]. In addition to clearance by phagocytosis, some ApoVs can be metabolized from the integumentary skin and hair follicles [35].

At present, specific markers to identify ApoVs are controversial. All ApoVs express common EV biomarkers, such as CD9, CD63, and CD81 [36]. During apoptosis, PS translocates to the outer leaflet of the lipid layer and binds to Annexin V, making Annexin V the most widely used marker of ApoVs [37]. In addition, oxidation of the membrane surface provides opportunities for binding sites of thrombospondin (TSP) or the complement protein, C3b, which are then recognized by phagocyte receptors [38]. Therefore, Annexin V, TSP, and C3b are well-accepted markers of ApoVs [39]. Our group recently identified that MSC-derived ApoVs (MSC-ApoVs) potentially contain 13 specific biomarkers (Fas, Integrin alpha-5, Syntaxin-4, CD44, RhoA, Caveolin-1, Cavin1, Rab-5C, ribosomal protein S25 (RPS25), Lamin B1, voltage-dependent anion channel 2 (VDAC-2), Calnexin, and Calreticulin) through diverse proteomic analyses and Western blotting. Moreover, syntenin-1 was also confirmed as an exclusive biomarker for ApoVs [21]. Specific markers mean preferentially loaded and abundant proteins in ApoVs compared with other EVs, indicating that they might be involved in important events of ApoVs, such as biogenesis, material transport, or biological functions. For instance, calreticulin, one of the specific markers, acts as a crucial ‘eat-me’ signal mediating ApoV efferocytosis and macrophage regulatory effects, which ameliorate type 2 diabetes [40]. Fas inside ApoVs mediates the direct contact between ApoVs and platelets, and subsequently activates platelet aggregation, thereby rescuing blood clotting disorders [21]. Exposed PS on ApoVs mediates the interaction with T cells to disrupt proximal T cell receptor signaling transduction, which then ameliorates inflammation and joint erosion in murine arthritis [41].

The components of ApoVs mainly depend on the identity of their original cells. In most studies, ApoVs carry cargoes including micronuclei, chromatin remnants, cytosol portions, degraded proteins, DNA fragments, or even intact organelles [24,42]. ApoVs derived from various cell types contain different cargoes and exhibit distinct functions [43]. For example, MSCs can engulf ApoVs and reuse MSC-ApoV-derived ubiquitin ligase RNF146 and miR-328-3p to maintain bone homeostasis [44]. Glioblastoma cell isolated-ApoVs, enriched with various components of spliceosomes, can alter RNA splicing in recipient cells and accelerate their therapeutic resistance and aggressive migratory phenotype [45]. Interestingly, DNA and RNA could not be packed into ApoVs of HL-60 cells simultaneously. Over 90% of ApoVs containing RNA had no detectable DNA and vice versa [46]. Moreover, some ApoVs have no nucleic acid materials. Nuclear contents are absent in ApoBDs derived from monocytes via a ‘beads-on-a-string’ protrusion [31]. This evidence suggests that some bioactive molecules might enter ApoVs via tropism. The molecular composition of EVs is a key focus in the field because they carry many functional proteins [47]. Studies have demonstrated that MSC-ApoVs are enriched with numerous functional proteins that are highly related to cellular behavior, cellular metabolism, and cellular transport, as well as the regulation of various diseases [21,40]. Further research is required to confirm the specific functions of enriched functional proteins in ApoVs.

### 2.2. Small EVs

The most popular subspecies of EVs are sEVs, which range in size from 30 to 150 nm in diameter and have a typical cup-shaped morphology under an electron microscope [48]. Zabeo et al. classified sEVs into nine different groups in terms of their shape: single vesicle, double vesicle, triple vesicle or more, small double vesicle, oval vesicle, small tubule, large tubule, incomplete vesicle, and pleomorphic vesicle [49]. In 1981, sEVs were first observed as plasma membrane-derived vesicles with 5′ nucleotide enzyme activity [50]. Thereafter, numerous studies reported that they can be produced from almost all cell types (e.g., human umbilical vein endothelial cells, reticulocytes, immune cells, macrophages, dendritic cells (DCs), natural killer (NK) cells, stem cells, fibroblasts, endothelial cells, epithelial cells, and neuronal cells) and can be found in body fluids (e.g., blood, breast milk, saliva, semen, and urine), as reviewed in [51]. sEVs are enriched with biologically active materials, thereby acting as biological messengers for intra- and inter-cellular communication.

The biogenesis of sEVs can be divided into three key stages: endocytosis, multivesicular body development, and release. First, the inward budding of the cell plasma membrane forms endocytic vesicles, which fuse with each other to arrange early sorting endosomes (ESEs). Later, ESEs develop into late-sorting endosomes (LSEs). Invagination of LSEs membranes results in the formation of intraluminal vesicles (ILVs) within large multivesicular bodies (MVBs). Finally, sEVs are released into the extracellular space upon the fusion of MVBs with the plasma membrane [52]. Importantly, MVBs are generated through endosomal sorting complexes required for transport (ESCRT)-dependent and ESCRT-independent mechanisms. The ESCRT-dependent pathway is mediated by a set of ESCRT complexes (ESCRT-0, ESCRT-I, ESCRT-II, and ESCRT-III) that participate in recruiting cargo and forming the inward invagination of the late endosomal membrane [53]. ESCRT-0, ESCRT-I, and ESCRT-II can recognize and load ubiquitinated cargo into the lumen of endosomes. ESCRT-II proteins are responsible for activating the assembly of ESCRT-III, which recruits related proteins to orchestrate the formation of ILVs in MVBs. The ESCRT-independent pathway can also regulate the inward budding of MVBs through sphingomyelinase hydrolysis and ceramide formation by neutral sphingomyelinase 2 [54]. Ceramide induces the spontaneous membrane germination of the MVBs to form ILVs, which are released into the extracellular space as sEVs [6].

The tetraspanins CD9, CD63, and CD81 have been traditionally used as biomarkers of sEVs, and have even been applied to purify and define the molecular characteristics of sEVs [20]. However, Kugeratski’s study reported the heterogeneous presence and varied abundance of CD9, CD63, and CD81 in sEVs, suggesting the limited function of these tetraspanins as biomarkers for sEVs derived from different cell types [36]. Moreover, the authors checked the expression of other frequently used sEV biomarkers, such as flotillin-1, flotillin-2, heat shock protein (HSP)70, HSP90, ALG-2-interacting protein X (ALIX), and tumor susceptibility 101 (TSG101). The results showed that only ALIX and TSG101 were ubiquitous and abundant in the sEVs from all 14 cell lines analyzed. A cohort of 22 proteins that are consistently enriched in sEVs could potentially serve as biomarkers of sEVs, such as integrin subunit beta 1 (ITGB1), galectin 3 binding protein (LGALS3BP), solute carrier family 3 member 2 (SLC3A2), ALIX, CD47, and TSG101. Notably, syntenin-1 is the most abundant protein in sEVs and is a putative universal biomarker candidate. Furthermore, 15 low-abundance proteins, including the frequently used exclusion marker calnexin, were validated as potential exclusion markers of sEVs, which are found in the nucleus, cytoplasm, mitochondria, endoplasmic reticulum (ER), and Golgi [36]. The enriched proteins as potential biomarkers of sEVs may be involved in key aspects of sEV biology. For example, ALIX, TSG101, and syntenin-1 are implicated in the biogenesis of exosomes and endosomal trafficking [55,56]. CD47 can protect cells and sEVs from phagocytosis, suggesting the possible mechanism by which sEVs escape phagocytic clearance [57]. ITGB1 can work synergistically with other proteins in cellular targeting and the internalization of exosomes [58].

Current data from different studies indicate that sEVs contain various substances, including proteins, lipids, nucleic acids, and other bioactive molecules. The components of sEVs vary widely based on the cell of origin, which dominates their functions [5]. sEVs contain a large variety of proteins with varied functions, such as tetraspanins (CD9, CD63, CD81, and CD82) that participate in cell penetration, invasion, and fusion events; heat shock proteins (HSP70, HSP90), which are associated with antigen binding and presentation; MVB formation proteins (ALIX, TSG101) that are engaged in exosome release; and other proteins (e.g., Annexins and Rab GTPase) accounting for membrane transport and fusion [59]. sEVs also carry different patterns of RNAs, including microRNAs (miRNAs), long non-coding RNAs (lncRNAs), piwi-interacting RNAs, transfer RNAs, small nuclear RNAs, and small nucleolar RNAs, which can be incorporated into recipient cells [60,61]. Although small amounts of DNA have been reported to be found in sEVs, some studies suggest that sEVs from human cell lines and serum do not contain DNA [7,20]. This contentious issue requires further quantitative studies. The lipid components of sEVs include PS, phosphatidic acid, cholesterol, sphingomyelin (SM), arachidonic acid, other fatty acids, prostaglandins, and leukotrienes, which are involved in the stability and structural rigidity of sEVs [62]. As mediators of cell–cell communication, sEVs play an important role in both physiological and pathological processes. Moreover, sEVs can carry specific therapeutic drugs and a wide range of elements, including lipids, RNAs, and signals via the transfer of proteins into recipient cells or tissues, making sEVs a promising diagnostic biomarker and therapeutic tool [7,59].

### 2.3. Microvesicles

Microvesicles, also called microparticles, ectosomes, or shedding vesicles, are irregular-shape vesicles released from the surface membranes of cells, with a size ranging from 100 to 1000 nm. MVs were first described as “platelet-like activity” in 1955 and later as “platelet dust” in 1967. Research progress on their composition, origin, and roles in physiological and pathological conditions led to these particles gradually becoming known as “microvesicles” [63]. MVs are produced and shed from almost all cell types [64]. Recently, research has shown that MVs not only serve as therapeutic agents but can also be utilized as drug-delivery vesicles [64]. Based on the genetic manipulation status of the original cells and bioactive materials, Agrahari et al. categorized MVs into the following subtypes: (i) native MVs derived from unmodified cells; (ii) MVs derived from genetically modified cells, but free of transgene products; (iii) MVs derived from genetically modified cells and containing transgene products; and (iv) native MVs used to deliver drugs or other molecules, such as RNA [65].

MVs are formed from cells under physiological conditions, such as cell growth, while the release of MVs is enhanced when cells are activated because of cell injury, proinflammatory stimulants, hypoxia, oxidative stress, or shear stress [66]. MVs are produced through the outward budding of the cell plasma membrane, which detaches and becomes a vesicle. However, the mechanisms by which MVs develop and bud from cell membranes are still largely unknown. Generally, the process is initiated by Ca^2+^ released from the ER, which activates several Ca^2+^-dependent enzymes. This causes the cytoskeleton to change shape and the phospholipid asymmetry to flip, thereby translocating PS to the cell membrane. Subsequently, following actomyosin contraction, the outward budding of the membrane splits and MVs are released from the cell surface into the extracellular space [67]. Membrane budding is regulated by phospholipid redistribution, together with Rho-kinase-mediated myosin light chain phosphorylation and the contractile machinery, to allow for vesicle pinching and detachment [6].

According to the biogenesis of MVs, the membrane molecules on MVs rely on their parent cells, providing specific molecules as biomarkers for MV identification, such as externalized PS, CD9, ADP ribosylation factor 6 (ARF6), TSG101, and CD63 [65,67]. However, these biomarkers are unable to distinguish MVs from other EVs, thus further research is needed in this area. It has been demonstrated that small EVs with CD9 and CD81 (but little CD63) are MVs, while small EVs with late endosomal proteins and CD63 (but little CD9) might qualify as exosomes [68]. Jeppesen et al. reported that annexin A1 is a novel and specific marker for MVs [20]. Actinin-4 is also a potential specific marker for MVs [69]. The possible unique functions of these specific markers require more research.

MVs carry plasma membrane receptors, proteins (including cytokines, chemokines, and biomolecules), lipids, carbohydrates, and genetic material (including mRNA and miRNAs) [66]. The internal compositions are decided by a wide range of factors, such as the parent cell, the microenvironment, and the stimulation conditions during their release [67]. For instance, MVs derived from endothelial cells with starvation or tumor necrosis factor-alpha (TNF-α) pre-treatment exhibited opposite functions in regulating the expression of intercellular adhesion molecule 1 (ICAM-1), which suggested the high heterogeneity of MVs [67]. Researchers found 112 different proteins and 50 different metabolites between sEVs and MVs from human plasma, indicating the distinct characteristics of MVs [69]. miRNAs encapsulated in MVs can be transported into target cells to silence target genes, thereby influencing recipient cell function [70]. For example, platelet-derived MVs can effectively deliver miR-223 into human lung cancer cells via erythrocyte membrane protein band 4.1 like 3 (EPB41L3), which promotes tumor invasion [71]. Therefore, cell-secreted miRNAs in MVs can serve as a novel class of signaling molecules to mediate distant intercellular communication. Furthermore, it also indicated that components of MVs can be changed by modification of the parent cells. Common modification methods include incubation and electroporation [67].

## 3. Mechanisms of EVs in Tissue Regeneration

The targeted delivery of EV cargoes to recipient cells plays essential roles in intercellular signal transduction and tissue homeostasis. EV-mediated cell-to-cell communication is regulated by three primary mechanisms: receptor-ligand interactions, direct plasma membrane fusion, and internalization into recipient cells [72]. The regenerative potential of EVs is mainly attributed to their direct impact on target cells in the regulation of cell apoptosis, proliferation, and differentiation, as well as their indirect effects on enhancing angiogenesis and modulating the immune microenvironment. Multiple lines of evidence have confirmed EVs as potential mediators of regenerative medicine in various tissues, such as the liver, heart, cardiovascular, skin, bone, and brain [73]. The possible mechanisms of EVs in tissue regeneration will be discussed as shown in Figure 3.

To restore tissue function, the number of cells has to be increased. EVs have been proven to have a direct impact on the proliferation, migration, and differentiation of certain specific cells. The various components of EVs can be involved in this process. Liu et al. demonstrated that impaired MSCs can reuse EV-derived ubiquitin ligase RNF146 and miR-328-3p to activate the Wnt/β-catenin pathway, which recovered their proliferation and differentiation capability to promote mineralized nodule formation [44]. EV-derived lncRNA KLF3-AS1 and H19 have been reported to enhance chondrocyte proliferation and suppress chondrocyte apoptosis, playing an important role in cartilage regeneration [74,75]. Another study showed that donor EVs can transfer Fas to recipient defective bone marrow mesenchymal stem cells (BMMSCs) to regulate the miR-29b/DNA methyltransferase 1/Notch epigenetic pathway, thereby enhancing the osteogenesis of recipient BMMSCs [76]. Proteome analysis of MSC EVs revealed the presence of proteins that participate in cell proliferation, adhesion, migration, and morphogenesis [77].

EVs also promote vascularization, which is an indispensable part of the regenerative microenvironment, via improved diffusion of oxygen and nutrients. Increasing evidence shows that EVs derived from MSCs can effectively revitalize angiogenesis in vitro and in vivo through the transfer of miRNAs [78]. For example, miR-21 isolated from umbilical MSCs is a potential intercellular messenger to promote angiogenesis by upregulating the NOTCH1/delta-like canonical notch ligand 4 (DLL4) pathway, exhibiting a promising EVs-based strategy to repair large bone defects [79]. miR-210 and miR-126 of MSCs have been reported to promote angiogenesis, improve cardiac function, and decrease the infarct area [80]. Proteins in EVs also contribute to angiogenesis during tissue repair and reconstruction. Anderson et al. identified that MSC-derived EVs contain several paracrine effectors of angiogenesis, including platelet-derived growth factor, epidermal growth factor, fibroblast growth factor, and most notably, nuclear factor-kappa B (NF-κB) signaling pathway proteins. NF-κB signaling was demonstrated as a key mediator of MSC-EVs-induced angiogenesis in endothelial cells. The abundance of these paracrine effectors was increased in MSCs exposed to ischemic tissue-simulated conditions, suggesting EVs’ potential for the treatment of ischemic tissue-related diseases [81].

Tissue injury is often accompanied by an immune response and the quality of injured tissue repair is highly affected by the immune response. Therefore, the immune response plays a significant role during the complicated process of tissue repair and regeneration. A recent review has summarized the immunomodulation capability of EVs in tissue regeneration [82]. EVs, including sEVs, MVs, and ApoVs, exert immunoregulatory effects on multiple immune cells, such as T cells, macrophages, dendritic cells, and neutrophils, by transferring miRNAs or carrying proteins and cytokines. As the most-targeted cells, macrophages play a key role in mediating tissue repair and regeneration by transforming from a proinflammatory subtype (M1) to a beneficial type (M2), which induces the progress from the inflammatory stage to the tissue repair stage [83]. For example, in a murine model of lung injury, EV treatment alleviated symptoms through increased expression of miR-27a-3p and induced the polarization of M2 macrophages via the NF-κB1 signaling pathway [84]. In addition to macrophage polarization, BMMSC-derived EVs could also inhibit M1 macrophage activation and localized neutrophil infiltration in a rat cerebral hemorrhage model [85].

EVs also function as carriers of growth factors and cytokines and, thus, facilitate tissue regeneration. Kou et al. have revealed that gingival MSC-derived EVs contain a significantly higher amount of an IL-1 receptor antagonist, a natural inhibitor of the proinflammatory cytokine IL-1, and promote accelerated wound healing [86]. It has been reported that EVs are involved in several well-characterized signaling pathways, including MAPK, Wnt/β-catenin, PI3K/Akt, Notch, TGFβ/SMADs, STAT, and Hedgehog signaling [73], which is related to their potential in tissue regeneration. To date, the exact mechanisms by which EVs contribute to tissue regeneration remain to be clarified.

## 4. EVs in Pulp Regeneration

EVs play a critical role in multiple tissue regeneration, including that of dental pulp [17]. Regenerative endodontics comprises several aspects, including pulp revascularization, pulpal tissue regeneration, dentin formation, and neurological recovery, which involves the migration, proliferation, and differentiation of vascular endothelial cells and dental stem cells, such as dental pulp stem cells (DPSCs), periodontal ligament stem cells (PDLSCs), gingival mesenchymal stem cells (GMSCs), dental follicle stem cells (DFSCs), stem cells from human exfoliated deciduous teeth (SHED), and stem cells from apical papilla (SCAP). The potential functions of EVs, mostly derived from dental stem cells, in endodontic regeneration will be discussed below and summarized in Table 1 and Figure 4.

### 4.1. Pulp Vascularization

Pulp vascularization is fundamental for functional pulp regeneration because sprouting angiogenesis is the predominant process during pulp regeneration and therapeutic processes [87]. The key point for successful angiogenesis during pulp regeneration is to provide an adequate blood supply. EVs derived from various stem cells have been reported to promote angiogenesis by increasing endothelial cell viability, proliferation, migration, tube formation, and ameliorating damaged endothelial cells (ECs) [88]. Interestingly, EVs from dental stem cells exhibited a high potential for pulp vascularization [16]. Xian et al. revealed that dental pulp cell-derived EVs are capable of facilitating human umbilical vein endothelial cell (HUVEC) proliferation, tube formation, and proangiogenic factor expression, including fibroblast growth factor 2 (FGF-2), vascular endothelial growth factor A (VEGFA), kinase insert domain receptor (KDR), and matrix metalloproteinase 9 (MMP-9), indicating vital roles in angiogenesis. Moreover, p38 MAPK signaling inhibition enhanced EV-induced tube formation, suggesting that p38 MAPK signaling participates in EV-mediated angiogenesis [89]. Using a tooth fragment model in immunocompromised mice, Wu et al. showed that SHED aggregate-derived EVs (SA-EVs) effectively improved pulp tissue regeneration and angiogenesis in vivo via promoting SHED endothelial differentiation and enhancing the angiogenesis of HUVECs. Mechanistically, SA-EV-transferred-miR-26a improved angiogenesis both in SHED and HUVECs by regulating the TGFβ/SMAD2/3 pathway, which contributes to pulp regeneration [90].

Apoptosis is an autonomously regulated programmed cell death and used to be considered a passive phenomenon, whereas recent studies suggest that apoptosis has an important role in modulating tissue homeostasis and regeneration [44,91,92]. Cells undergo apoptosis after implantation in an ischemic-hypoxic environment; however, the roles of EVs released by apoptotic cells are largely unknown. Recently, Li et al. demonstrated that ApoVs released by SHED can be internalized by endothelial cells to augment their angiogenic capacities, including their proliferation, migration, differentiation, and secretion. ApoV-shuttled mitochondrial Tu translation elongation factor could modulate the activation of endogenous ECs through the transcription factor EB-autophagy pathway. In a beagle model, endogenous ECs were recruited by ApoVs to promote the formation of blood vessel-rich dental-pulp-like tissue [92]. The experimental data revealed the significance of apoptosis in tissue regeneration and demonstrated the potential of using ApoVs to promote angiogenesis in pulp regeneration.

Notably, EVs derived from cells in an inflammatory state possess higher proangiogenic potential. EVs secreted by DPSCs isolated from periodontally diseased teeth exerted enhanced potential to promote the angiogenesis of ECs in vitro and were better able to accelerate cutaneous wound healing and promote vascularization in vivo, compared with those from periodontally healthy teeth [93]. Similarly, the number of EVs derived from PDLSCs was enhanced by inflammation, while EVs could promote the angiogenesis of HUVECs by mediating VEGFA transfer via miR-17-5p [94]. Furthermore, EVs originating from lipopolysaccharide (LPS)-preconditioned DPSCs had a more significant effect in modulating BMMSC proliferation, migration, angiogenesis, and differentiation compared with EVs isolated from normal cells. The results demonstrated that EVs released by DPSCs in a mild inflammatory microenvironment are capable of facilitating the regeneration of dental pulp through functional healing instead of scar healing, which has potential applications in regenerative endodontics [95]. The above findings confirmed a generally accepted conclusion that submitting cells to suitable environmental stress factors, such as radiation, oxidative stress, hypoxia, or inflammation, could increase the pro-angiogenic potential of their EVs. The underlying angiogenesis potential of EVs stimulated by stress factors needs to be explored for dental pulp regeneration.

### 4.2. Nerve Regeneration and Neural Repair

Neuralization of the damaged tissue is critical for the regeneration of functional dental pulp because nerves perform sensory functions in the pulp and are responsible for responding to external stimuli [96]. The ideal regenerative endodontics also involve nerve regeneration and neural recovery. Dental stem cells, originating from cranial neural crests, are suitable for the induction of neural differentiation during pulp regeneration procedures. Previous studies have shown that transplantation of DPSCs can promote neurite extension and neuron growth during pulp regeneration, but little is known about the neurogenesis capability of EVs for pulp regeneration [11,97]. EVs have been reported to improve neural recovery and induce neurogenesis in some neural diseases [98]. Schwann cells (SCs) play a vital role in the support, maintenance, and regeneration of nerve fibers in dental pulp [99]. Recently, Wang et al. revealed that EVs derived from SCs (SC-EVs) enhanced neurite outgrowth and neuron migration of rat dorsal root ganglia explants after coculture, suggesting that SC-EVs are able to promote neural regeneration and might facilitate the regeneration of functional nerve fibers in dental pulp tissue [98]. To investigate the role of EVs from the dental pulp tissue (DPT-EVs), Chen et al. built an in vivo “cell homing” model by filling the root canal of swine teeth with a mixture of treated dental matrix and DPT-EV-laden scaffolds. After 8 weeks of subcutaneous implantation into immunodeficient nude mice, the results showed that the DPT-EVs promoted the neurogenetic differentiation of SCAP by expressing neurogenetic markers MBP101 and neurofilament protein (NF200), indicating the potential of DPT-EVs in nerve regeneration [96].

Recent studies suggested the positive effects of EVs on neurogenesis, indicating that EVs are a potential biomimetic tool during pulp regeneration. However, the underlying mechanism and role of EVs during pulp nerve regeneration and neural recovery remains to be clarified.

### 4.3. Dentin-Pulp Complex Regeneration

An ideal outcome of dental pulp regeneration also requires recruited or implanted MSCs’ multipotency to form the dentin-pulp complex, which is the frontier of reparative dentin formation and the foundation of ideal pulp regeneration [100]. On this basis, the biological function of dental pulp is subsequently reconstructed. EVs have been proven to improve cell differentiation and can be used as a powerful tool for pulp regeneration [101].

Endocytosis of dental pulp cell-derived EVs by DPSCs and BMMSCs induced the odontogenic differentiation of both cells by triggering the increased expression of DSPP (encoding dentin sialophosphoprotein). When tested in vivo, EVs also triggered the regeneration of dental pulp-like tissue. Interestingly, EVs derived from cells under odontogenic conditions are more potent in inducing lineage-specific differentiation of DPSCs, suggesting that the source and state of EVs are critical for their therapeutic potential [102]. Hu et al. further investigated the underlying mechanism via microRNA sequencing and pathway analysis. They found that 28 microRNAs in EVs extracted from DPSCs cultured in odontogenic medium were significantly changed compared with those in regular medium. In addition, conditioned EV treatment or transfection of miR-27a-5p, which was upregulated in EVs under odontogenic induction, both enhanced TGFβ signaling and promoted odontogenic differentiation of DPSCs. Further experimental data verified that the EV-encapsulated miR-27a-5p promoted odontogenic differentiation through the TGFβ1/SMADs signaling pathway by downregulating latent TGF-β-binding protein 1 (LTBP1) [103]. Recently, EVs derived from SCAPs were observed to be taken up by BMMSCs and markedly improved their specific dentinogenesis in root fragments transplanted into nude mice [104]. A previous study reported that EVs derived from a Hertwig’s epithelial root sheath (HERS) cell line could promote the odontogenic differentiation of DPSCs and enhance dentin-pulp complex formation [105]. All these findings indicated that the use of EVs from dental stem cells could constitute a potential therapeutic approach for dentine-pulp complex regeneration in regenerative endodontic procedures.

### 4.4. Immunomodulatory Properties in Pulp Regeneration

The continuous production of inflammatory cytokines in pulp would maintain inflammatory reactions, eventually resulting in dental pulp necrosis. Therefore, suitable inflammation conditions are necessary for pulp regeneration. EVs inherit the function of their parent cells and have lower immunogenicity; therefore, they play important roles in immune regulation and tissue regeneration [106]. For example, EVs derived from human GMSCs (GMSC-EVs) could inhibit the inflammatory response of PDLSCs by regulating the expression of NF-κB signaling and Wnt5a, which restored the regenerative potential of PDLSCs and promoted periodontal tissue regeneration in patients with periodontitis [107].

Macrophages are the most abundant immune cells in pulp, acting as the critical regulators of inflammation-related diseases, such as pulpitis. They are crucial for inflammatory pulp regeneration because their interactions with pulpal inflammation can create a regulatory microenvironment for the odontogenesis of stem cells [108]. Zheng et al. clarified that microRNA-enriched EVs derived from DPSCs (DPSC-EVs) possess odonto-immunomodulatory properties by switching macrophages from the M1 to the M2 phenotype to enhance the odontogenesis of DPSCs. Mechanistically, miR-125a-3p was significantly upregulated in DPSC-EVs, which was proven to mediate macrophage phenotype switching via inhibition of NF-κΒ and Toll like receptor (TLR) signaling. Moreover, DPSC-EVs and the encapsulated miR-125a-3p both enhanced bone morphogenetic protein 2 (BMP2) release in macrophages, promoting the odontogenesis of DPSCs through BMP2 pathway activation [109]. The immune microenvironment is critical during inflammatory dental pulp regeneration, especially T cell-based immune functions. A previous study showed that SCAP-EVs could enhance Treg conversion and effectively alleviate inflammation in the dental pulp of rats, indicating that SCAP-EVs can modulate the local immune microenvironment to support tissue regeneration [110]. Recent studies have shown that EVs can alleviate inflammation to promote pulp regeneration in vitro and in vivo (Table 1); however, the detailed mechanism of how EVs regulate the immune balance during pulp regeneration requires further investigation.

**Table 1 pharmaceutics-15-00282-t001:** Therapeutic effects of EVs in dental pulp regeneration.

Origin of EVs	Types of EVs	Study Models	Key Functions/Potential Molecular Mechanism	References
Dental pulp stem cells (DPSCs)	sEVs	In vitro and in vivo (teeth root slices implanted subcutaneously into nude mice dorsum)	Induced lineage-specific differentiation of stem cells/Triggered the P38 MAPK pathway	Huang et al., 2016 [102]
Dental pulp cells	sEVs	In vitro	Promoted angiogenesis/Enhanced tubular morphogenesis via p38 MAPK signaling inhibition	Xian et al., 2018 [89]
DPSCs	sEVs	In vitro and in vivo (dental pulp capping in SD rat teeth)	Enhanced odontogenesis by switching macrophages toward pro-healing M2 cells/Promoted odontogenesis in DPSCs through BMP2 pathway activation	Zheng et al., 2020 [109]
Hertwig’s epithelial root sheath (HERS) cells	sEVs-like vesicles	In vitro and in vivo (transplantation in renal capsule of rat; teeth root slices transplanted subcutaneously into nude mice dorsum)	Triggered regeneration of dental pulp-dentin-like tissue comprised of hard (reparative dentin-like tissue) and soft (blood vessels and neurons) tissue/Endocytosis of sEVs triggered the activation of P38 MAPK pathway	Zhang et al., 2020 [105]
DPSCs	EVs	In vitro	Promoted angiogenesis in an injectable hydrogel in vitro	Zhang et al., 2020 [111]
Stem cells from apical papilla (SCAPs)	sEVs	In vitro and in vivo (teeth root fragments implanted subcutaneously into nude mice dorsum)	Endocytosed by bone marrow-derived mesenchymal stem cells (BMMSCs) and significantly improved their specific dentinogenesis	Zhuang et al., 2020 [104]
Dental pulp cells	sEVs	In vitro	Induced the recruitment and proliferation of human mesenchymal stem cells	Ivica et al., 2020 [112]
Stem cells from human exfoliated deciduous teeth (SHED)	sEVs	In vitro and in vivo (teeth root fragments implanted subcutaneously into mice dorsum)	Shuttled miR-26a to promote angiogenesis via TGF- β/SMAD2/3 signaling	Wu et al., 2021 [90]
DPSCs	sEVs	In vitro	Lipopolysaccharide (LPS)-stimulated sEVs displayed a better ability on regulating SCs migration and odontogenic differentiation than normal sEVs	Li et al., 2021 [113]
DPSCs	sEVs	In vitro	LPS stimulated sEVs (LPS-sEVs) showed better proangiogenic potential of HUVECs compared with control sEVs/The expression of miR-146a-5p, miR-92b-5p, miR-218-5p, miR-23b-5p, miR-2110, miR-27a-5p, and miR-200b-3p was increased in the LPS-sEVs/The expression of miR-223-3p, miR-1246, and miR-494-3p was decreased in the LPS-sEVs	Huang et al., 2021 [114]
Dental pulp cells	sEVs	In vitro and in vivo (dental pulp capping in minipig teeth)	sEVs-treated dentin matrix promoted the formation of continuous reparative dentin	Wen et al., 2021 [115]
Dental pulp tissue and cells	sEVs	In vitro and in vivo (swine teeth implanted subcutaneously into mice dorsum)	Recruited SCAPs to regenerate connective tissue similar to natural dental pulp	Chen et al., 2022 [96]
Platelet	sEVs	In vitro	5% thrombin-activated platelet-derived sEVs had a high potential to induce dental pulp regeneration	Bagio et al., 2022 [116]
Dental pulp cells	EVs	In vitro and in vivo (subcutaneous transplantation in nude mice)	Regulated cellular NFIC level in SCAPs to promote the proliferation, migration of SCAPs, and dentinogenesis	Yang et al., 2022 [117]
SHED	ApoVs	In vitro and in vivo (teeth fragments implanted subcutaneously into nude mice dorsum; orthotopic model of beagle dog)	Recruited endogenous endothelial cells (ECs) and facilitated the formation of dental-pulp-like tissue rich in blood vessels/ApoVs-carried mitochondrial Tu translation elongation factor modulated the angiogenic activation of ECs through the transcription factor EB-autophagy pathway	Li et al., 2022 [92]

## 5. EVs in Periodontal Regeneration

The American Academy of Periodontology defined periodontal regeneration as the formation of a new cementum, alveolar bone, and a functional periodontal ligament over a previously diseased root surface [118]. Tissue engineering and cell-based therapies have been considered novel alternatives to overcome the limitations of existing therapies, because of the reported functions of stem cells in promoting and regulating tissue regeneration [15,119]. MSCs participate in the regeneration of defective or damaged periodontal tissues because of their high proliferation, multipotency, paracrine effects, and immune regulation [120]. As an important part of the stem cell secretome, EVs have been demonstrated to participate in periodontal tissue repair and regeneration.

In 2019, Chew et al. investigated the therapeutic effects of MSC-EV-loaded collagen sponges to regenerate surgically created periodontal intrabony defects in an immunocompetent rat model. The data showed that EV-treated rats repaired the defects more efficiently with the regeneration of periodontal tissues, including newly-formed bone and PDL, possibly via increasing PDLSCs migration and proliferation [121]. PDLSCs, residing in the perivascular space of the periodontium possess MSC potentials and are a therapeutic target for periodontal regeneration [122]. A later study reported that EVs secreted from healthy PDLSCs (h-PDLSC-EVs) promoted the osteogenic differentiation of PDLSCs derived from periodontitis tissue. h-PDLSC-EV treatment accelerated bone formation in the defect of alveolar bone in rat models of periodontitis. Mechanistically, h-PDLSC-EVs inhibited the over-activation of canonical Wnt signaling to recover the osteogenesis of inflammatory PDLSCs [123]. Recently, EVs isolated from DFSCs (DFSC-EVs) have also been demonstrated to improve periodontal tissue regeneration by promoting the proliferation, migration, and osteogenic differentiation of PDLSCs. The effect of DFSC-EVs might be partially induced by the activation of the p38 MAPK signaling pathway [124]. BMMSCs are crucial for desired bone regeneration. EVs derived from SHED (SHED-EVs) have been reported to directly promote osteogenesis of BMMSCs and suppress adipogenesis, thereby enhancing bone formation in a mouse model of periodontitis [125]. SHED-EVs also present a potential to mobilize naïve BMMSCs, suggesting their relevance in assisting bone regeneration [126]. Local administration of EVs to treat periodontitis has limitations, such as the short half-lives of EVs and their rapid diffusion away from the delivery site. To overcome these drawbacks, “dual delivery” microparticles have been designed, which not only facilitate the microenvironment-sensitive release of EVs by metalloproteinases at the affected site but also are loaded with antibiotics to suppress bacterial biofilm growth. The results showed that the one-time administration of immunomodulatory GMSC-EV-decorated microparticles led to a significant improvement in the regeneration of the damaged periodontal tissues [127].

Periodontitis is mainly caused by a host immune-inflammatory response to bacterial insult; therefore, it is critical to suppress the inflammatory immune microenvironments that mediate periodontal tissue damage for high-quality healing [14]. We have discussed that EVs derived from dental stem cells can modulate the local immune microenvironment to support pulp regeneration, which also exhibit their immunoregulation effects in periodontal regeneration. Zarubova et al. have shown that GMSC-EVs can reduce the secretion of pro-inflammatory cytokines by immune cells, inhibit T-cell activation, and induce the formation of Tregs in vitro. In a rat model of periodontal disease, GMSC-EVs led to a significant improvement in the regeneration of the damaged periodontal tissue [127]. DPSC-EVs can alleviate periodontitis-induced epithelial lesions and reduce alveolar bone loss in experimental mice by converting macrophages from a pro-inflammatory phenotype (M1) to an anti-inflammatory phenotype (M2). It was further clarified that miR-1246 within DPSC-EVs accounted for the therapeutic effect [128]. Research suggested that macrophages are also critically involved in the periodontal regeneration mediated by EVs, similar to their roles in pulp regeneration. Other research demonstrated that BMMSC-EVs suppressed the development of periodontitis and immune damage of periodontal tissue, partly attributed to the regulation of macrophage polarization, TGF-β1 expression, and osteoclast function [129]. In accordance with the response to an inflammatory environment in pulp regeneration, studies also reported that preconditioned EVs possess enhanced therapeutic potential in periodontal bone defects. GMSCs under TNF-α stimulation not only secreted increased amounts of EVs but also regulated inflammation by inducing M2 macrophage polarization. Moreover, EVs derived from TNF-α-preconditioned GMSCs greatly enhanced their therapeutic efficiency in reducing periodontal bone resorption and the number of osteoclasts in a mouse model of periodontitis, which was mediated by the upregulated miR-1260b within EVs targeting the Wnt5a-mediated receptor activator of nuclear factor kappa B ligand (RANKL) pathway and inhibiting its osteoclastogenic activity [130]. Similarly, Kang et al. reported that the TNF-α-preconditioned MSC-EVs possess enhanced immunomodulatory properties by suppressing M1 macrophages and increasing M2 macrophages in vitro, and reducing inflammation in vivo in a rat calvarial defect model. An analysis of EV miRNA composition revealed significant changes in anti-inflammatory miRNAs in the preconditioned MSC-EVs [131]. Taken together, these studies indicated the specific function of TNF-α-preconditioned EV miRNAs in the immunomodulatory control of periodontal and bone regeneration. Paradoxically, LPS-preconditioned EVs were beneficial to repair lost alveolar bone in the early stage of treatment and to maintaining the level of alveolar bone in the late stage of treatment in experimental periodontitis rats; however, the therapeutic potential was similar to the unstimulated EVs [132]. Further investigations are required to explain the discrepancy.

The above studies showed that EV-based therapies, especially dental EVs, are promising therapeutic approaches and have the potential to enhance the success of periodontal regeneration, as summarized in Table 2 and Figure 4. However, the clinical application of EVs is still challenging and further clinical investigations are needed to testify to the effects of EVs on patients.

## 6. Conclusions and Future Perspectives

EV-based therapies are emerging as novel cell-free therapeutic tools that provide advantages in avoiding ethical concerns and reducing risks of tumorigenesis, immunogenicity, and infections associated with direct cell therapy [135]. Knowledge related to EV biology, structure, contents, markers, and their mode of action mode has greatly expanded. In this review, we summarized recent studies of the applications of EVs in pulp and periodontal tissue regeneration. Based on these studies, EVs have the potential to regulate the immune microenvironment, boost angiogenesis, facilitate neural regeneration, and promote MSC proliferation and differentiation. It is well established that the properties of secreted EVs can be determined by their parent cells and the culture conditions. Current research indicates a great potential for EV application to improve the success and predictability of pulp and periodontal tissue regeneration therapies.

To date, over 140 EV-based clinical trials have been launched to treat various diseases (data from https://clinicaltrials.gov/ accessed on 29 December 2022), such as intracerebral hemorrhage, cancer, post-stroke dementia, alopecia, acute respiratory distress syndrome, sepsis, and COVID-19-related disease. However, a series of issues need to be resolved for their clinical application. First, scalable and reproducible methods to isolate and purify EVs are lacking. To date, a number of methods have been applied for EV isolation, such as ultracentrifugation (UC), density gradient centrifugation (DGC), filtration, size-exclusion chromatography (SEC), precipitation, microfluidics, and immunoaffinity techniques [136]. Although each method shows advantages as well as disadvantages, UC is the most commonly used method for EV isolation. As the current gold-standard technique for EV isolation, UC separates and concentrates EVs from other sample compositions by differential centrifugation based on their density [137]. The EV products by UC are prone to contamination because of the co-isolation of non-EV components, especially for biological fluids, such as serum and urine. Besides, high shear forces might induce EV aggregation or breakage, resulting in reduced biological activity. Ultracentrifugation is time-consuming and has low throughput, limiting its use to small-scale studies. Tangential flow filtration (TFF) combined with SEC has been recommended to produce EVs with high purity and biofunctions from large amounts of samples with minimal vesicle loss [138,139]. Recently, microfluidics technology has been applied to EV study, which enables the isolation, detection, characterization, and analysis of EVs on a single platform to simplify workflows without the need for large and expensive equipment [140]. Microfluidic separation techniques can be divided into four categories: active (acoustic field method, electric field method, and centrifugal method), passive (deterministic lateral displacement and viscoelastic-flow sorting), immunoaffinity-based, and combinative [141]. Microfluidics possesses significant advantages for EV isolation, such as low sample volume, time and cost efficiency, high throughput, and high size selectivity. However, this method takes time to fabricate microfluidic chips, fails to deal with large volume samples, requires a certain level of expertise, and is prone to clogging [136,142]. Thereby, the purity of the EVs used in different studies is questionable because standard operating procedures (SOPs) for EV production have not yet been established. No specific isolation technique has currently been accepted as suitable to separate ApoVs, sEVs, or MVs from the other subtypes of membrane-bound EVs. Even with characterization by EV-specific markers, the contribution of other types of EV cannot be totally excluded. SOPs are an urgent issue to promote the standardization of EV preparation and the isolation of high-quality EVs in bulk.

Second, storage conditions still need further study. The storage and recovery conditions of isolated EVs influence their characteristics, such as stability, amounts, aggregation, and bioactivities. A variety of factors during storage can affect the physiochemical and biological properties of EVs, including storage duration, temperature, relative humidity, or PH. All possible elements need to be determined to understand the degradation patterns and to define the limits of stability. Common storage methods of EVs include freezing at −80 °C and lyophilization. The freeze-thaw cycle during cryopreservation or the addition of lyoprotectants and excipients under freeze-drying also need to be carefully considered [143].

Another issue is the limited knowledge about the key mechanisms that mediate the functional cargos packaged into EVs and the transportation of EVs into targeted cells or tissues. There is a certain risk if it fails to be accurately regulated. Not all EVs have therapeutic functions. For instance, EVs have been reported to promote tumor occurrence and development by transmitting related miRNAs to tumor cells or activating signaling pathways [144]. Therefore, increasing our understanding of the mechanism for the sorting of critical cargos and selecting recipient cells might not only provide mechanistic insights into the role of EVs but also produce new tools to boost the therapeutic potential of EVs, especially for tissue repair and regeneration.

The last concern is about the safety guarantee for EV clinical application. As EVs carry complex cargos of proteins, lipids, and genetic materials, compositional analysis (e.g., comprehensive multi-omic studies) is necessary for checking unexpected contents, such as oncogenes and toxic molecules. The route, dosage, and frequency of administration also influence the safety of EVs. The systemic administration of EVs has been reported as the most common route of administration [145]. Pulp and periodontal defects are localized; therefore, local administration of EVs is more suitable for better control and fewer side effects of EV treatment. Before launching clinical trials, the effect of dose and frequency of the administration of EVs on toxicity and immunogenicity need to be evaluated in vivo. Accounts of the pharmacodynamics, pharmacokinetics, and potential off-target effects of EVs delivered via the proposed route of administration should be considered. In summary, continued efforts should be made to accelerate the clinical translation of EVs.

## Figures and Tables

**Figure 1 pharmaceutics-15-00282-f001:**
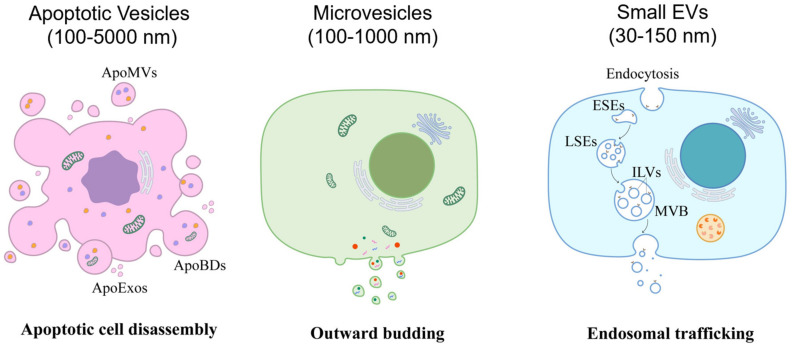
Graphic representation of the different types of extracellular vesicles (EVs). Generally, EVs are divided into apoptotic vesicles (ApoVs), microvesicles (MVs), and small EVs (sEVs) according to their biogenesis. ApoVs are released from disassembled apoptotic cells and are categorized as apoptotic bodies (ApoBDs), apoptotic microvesicles (ApoMVs), and apoptotic exosomes (ApoExos), based on their different sizes. MVs are generated via the direct outward budding of the plasma membrane. The biogenesis of sEVs comprises the formation of endocytic vesicles, early sorting endosomes (ESEs), late sorting endosomes (LSEs), and intraluminal vesicles (ILVs) within multivesicular bodies (MVBs), followed by release upon fusion of MVBs with the plasma membrane.

**Figure 2 pharmaceutics-15-00282-f002:**
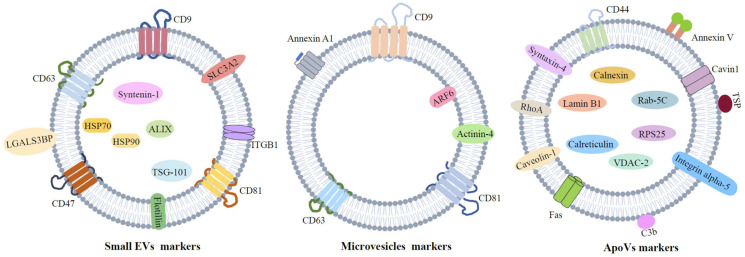
The markers of different kinds of EVs. The possible markers of sEVs, MVs, and ApoVs are briefly summarized.

**Figure 3 pharmaceutics-15-00282-f003:**
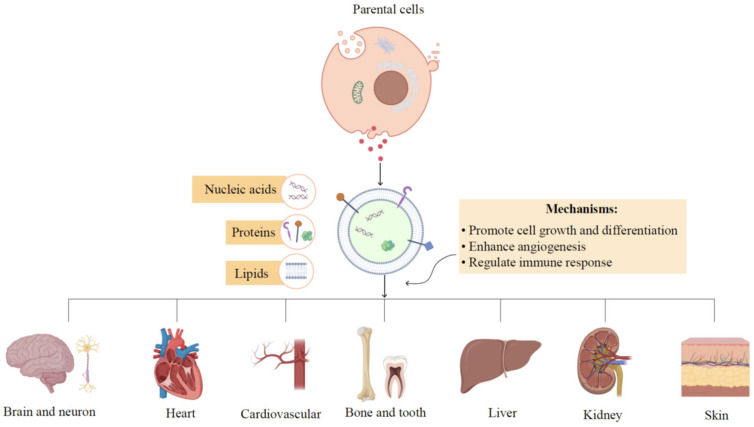
Therapeutic mechanisms of EVs in tissue regeneration. EVs derived from various parent cells contain nucleic acids, proteins, and lipids, which exhibit their functions in tissue regeneration. The regenerative potential of EVs is mainly attributed to their direct impacts on target cells in the regulation of cell apoptosis, proliferation, and differentiation, as well as their indirect effects on enhancing angiogenesis and modulating immune responses.

**Figure 4 pharmaceutics-15-00282-f004:**
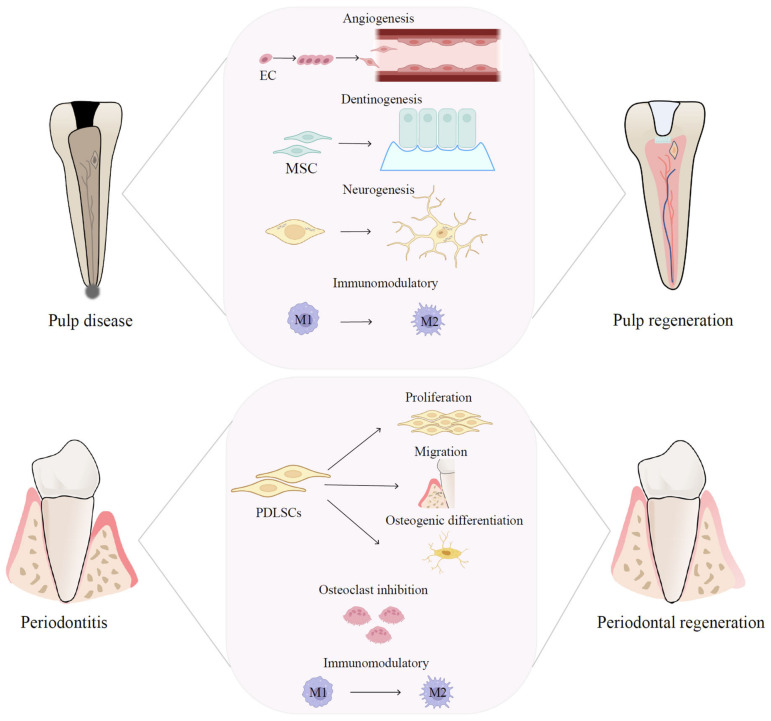
Therapeutic mechanisms of EVs in pulp and periodontal regeneration. EV-based therapies are promising therapeutic approaches for pulp and periodontal regeneration and have the potential to regulate the immune microenvironment, boost angiogenesis, facilitate neural regeneration, and promote MSC proliferation and differentiation.

**Table 2 pharmaceutics-15-00282-t002:** Therapeutic effects of EVs in periodontal regeneration.

Origin of EVs	Types of EVs	Study Models	Key Functions/Potential Molecular Mechanism	References
Adipose-derived stem cells	sEVs	In vitro and in vivo (ligature-induced periodontitis in rat)	Induced highly organized structures formation that was comparable to normal healthy periodontal tissue	Mohammed et al., 2018 [133]
Mesenchymal stem cells (MSCs)	sEVs	In vitro and in vivo (periodontal intrabony defect in rat)	Increased periodontal ligament (PDL) cell migration and proliferation through CD73-mediated adenosine receptor activation of pro-survival AKT and ERK signaling	Chew et al., 2019 [121]
Dental follicle cells (DFCs)	sEVs	In vitro and in vivo (ligature-induced periodontitis in rat)	Inhibited osteoclast formation and promoted periodontal regeneration via OPG/RANK/RANKL signaling pathway	Shi et al., 2020 [132]
DPSCs	sEVs	In vitro and in vivo (ligature-induced periodontitis in mouse)	Modulated macrophages from a pro-inflammatory phenotype to an anti-inflammatory phenotype in the periodontium of mice with periodontitis, which could be associated with miR-1246	Shen et al., 2020 [128]
SHED	sEVs	In vitro and in vivo (ligature-induced periodontitis +Porphyromonas gingivalis inoculation in mouse)	Directly promoted BMSCs osteogenesis, differentiation, and bone formation	Wei et al., 2020 [125]
BMMSCs	sEVs	In vitro and in vivo (ligature-induced periodontitis in rat)	Promoted the regeneration of periodontal tissues/Regulated the inflammatory immune response via the OPG–RANKL–RANK signaling pathway	Liu et al., 2021 [129]
Gingival mesenchymal stem cells (GMSCs)	sEVs	In vitro and in vivo (ligature-induced periodontitis in mouse)	Induced the resolution of inflammation and prevented bone loss in the periodontal tissue/Inhibited osteoclastogenesis via the Wnt5a-mediated RANKL pathway	Nakao et al., 2021 [130]
Periodontal ligament stem cells (PDLSCs)	sEVs	In vitro and in vivo (surgically created periodontal defect and ligature-induced periodontitis +LPS inoculation in rat)	Accelerated bone formation in the defect of alveolar bone in rat models of periodontitis/Recovered the osteogenic differentiation capacity of inflammatory PDLSCs via suppressing the over-activation of canonical Wnt signaling	Lei et al., 2022 [123]
Dental follicle stem cells (DFSCs)	sEVs	In vitro and in vivo (surgically created periodontal defect in rat)	Promoted periodontal tissue regeneration/Enhanced the proliferation, migration, and osteogenic differentiation of PDLSCs through the p38 MAPK signaling pathway	Ma et al., 2022 [124]
DFCs	MVs	In vitro and in vivo (surgically created periodontal defect in rat)	Strengthened alveolar bone regeneration through activating PLC/PKC/MAPK pathways	Yi et al., 2022 [134]

## Data Availability

Not applicable.

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
