# Peer review of "Extracellular Vesicles for Dental Pulp and Periodontal Regeneration"

_pharmaceutics, 2023, doi:10.3390/pharmaceutics15010282_

Round 1

Reviewer 1 Report

This review paper is well-organized and quite helpful to reader in this field. 

Author Response

This review paper is well-organized and quite helpful to reader in this field.

Response: We appreciate the reviewer’s positive comments.

Reviewer 2 Report

The review "Extracellular Vesicles for Dental Pulp and Periodontal Regeneration" is good one.

Major points

How it advance the current field, please make it clear

What are the literature selection and exculsion criteria for this review.

Why it is important

Please add some contents about the clinical translation

The first two figures are adding nothing

Table 1 is great and alligned with central theme of the review.

Please focus only dental regeneration and dental EVs

Author Response

The point-by-point response can be found in the attachment. 

Reviewer 3 Report

In this review article, Lai and colleagues reviewed extracellular vesicles for dental pulp and periodontal regeneration. The review article is well-organized and timely, covering many aspects of dental pulp and periodontal regeneration. The authors have also provided the challenges in the field in the conclusion section, which requires attention. I do recommend the article for publication after some enrichment. To further enrich the article, the authors are encouraged to:

1.       Discuss the role of microfluidics and nanotechnology on extracellular vesicles. Many microfluidics devices have been developed for extracellular vesicles for a variety of applications. Authors are encouraged to add some discussion on this concept, especially in conclusion, microfluidics has also been used for EV isolation.

2.       Also, the current gold standard technique for EV isolation must be mentioned and discussed.

3.       In section 2.1, apoptotic vesicles are defined as nanosized vesicles; however, these vesicles can be as large as 5 um, so the definition requires refinement.

4.       Proper illustrations are missing in the article. The article has only three figures, where one of them is about different types of EVS, the other one is about markers of EVs, and the last one is about the therapeutic mechanism of EVs. None of these indeed is about the topic of this article, which is dental pulp and periodontal regeneration. Authors are required to enrich the article in this aspect. 

Author Response

(The authors gave the same response as above.)
